# NaviFormer: A Deep Reinforcement Learning Transformer-like Model to Holistically Solve the Navigation Problem

## Abstract

Automatic path planning is a highly relevant research area with multiple applications, but it is usually solved by addressing either the (high-level) route planning problem (waypoint sequencing to achieve the final goal) or the (low-level) path planning problem (trajectory prediction between two waypoints avoiding collisions). However, real-world problems usually require simultaneous solutions to the route and path planning subproblems with a holistic and more efficient approach. In this paper, we introduce NaviFormer, a deep reinforcement learning model based on a Transformer architecture that solves the global navigation problem by predicting both high-level routes and low-level trajectories. To evaluate NaviFormer, several experiments have been conducted, including comparisons with other algorithms. Results show high competitive accuracy from NaviFormer since it can understand the constraints and difficulties of each high- and low-level planning and act consequently to improve the performance. Moreover, its superior computation speed proves its suitability for real-time applications.

## 1 Introduction

Planning tours for automated agents is a complex task with multiple applications, such as package delivery, transportation, autonomous vehicles, film-making, surveillance, search-and-rescue, exploration, military, and agriculture. The general idea is to generate routes for a given ground, aerial, or underwater unmanned vehicle that connect a set of waypoints/regions where the agent is expected to perform one or several tasks. Real-world scenarios usually impose constraints such as limited battery or fuel budget, which may prevent agents from visiting all regions; or static/dynamic obstacles, which may force to plan a new, typically suboptimal solution. The planning problem that models these real-world scenarios is usually divided into 2 subproblems: route planning and path planning.

Route planning consists of high-level planning to infer the best sequence of waypoints (route) without considering the exact trajectory between pairs of waypoints. It is frequently modeled as a Combinatorial Optimization Problem (COP) Mazyavkina et al. (2021), such as the Traveling Salesman Problem (TSP) Bellmore & Nemhauser (1968), which minimizes the distance of the route to visit every waypoint; the Orienteering Problem (OP) Golden et al. (1987), which is a variant of the TSP with time limitations; or the Capacitated Vehicle Routing Problem (CVRP) Dantzig & Ramser (1959), which substitutes the time constraint of the OP with a limited carrying capacity to comply with the demand of items in each region. The mentioned COPs do not consider obstacles, which allows assuming that every path is the straight line (Euclidean distance) connecting two waypoints.

Path planning Patle et al. (2019) focuses on finding the shortest low-level trajectory (path) defined by a discretized sequence of coordinates. Contrary to route planning, path planning considers static (prior knowledge) and/or moving (dynamically detected) obstacles that hinder the travel from a start to a goal waypoint. However, it is not involved in deciding the next node for a high-level objective.

In this paper, we propose a new and holistic definition of the problem, called Navigation Orienteering Problem (NOP). Previous works usually focus on either path planning Jin et al. (2023); Wu et al. (2023); Yu et al. (2022); Pehlivanoglu & Pehlivanoglu (2021) or route planning Kool et al. (2019); Fuertes et al. (2023); Ma et al. (2020); Ruano et al. (2017), and include the constraints that best fit the target application. On the contrary, we define a combination of both subproblems (see Figure

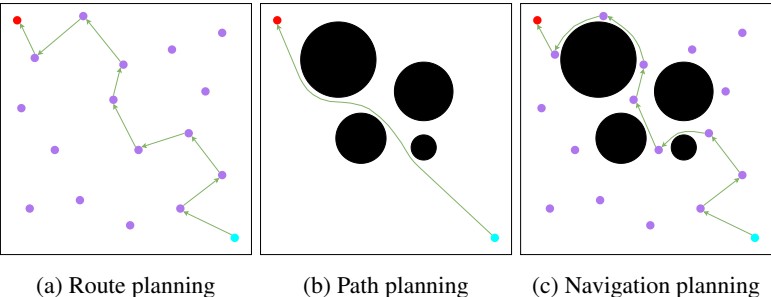

| (a) Route planning | (b) Path planning | (c) Navigation planning |

Figure 1: Example of problems: (a) Route planning - maximize number of visited regions (purple circles) from an initial to an end depot (cyan and red circles) within a time limit (like in the OP); (b) Path planning - find the shortest path to a goal region while avoiding obstacles (black circles); and (c) Navigation planning - holistic combination of the other two subproblems.

1). This definition includes basic constraints that would fit any application, such as the time limit (like in the OP), associated with the agent's fuel; the presence of obstacles, which exist in most of the real-world scenarios; and the necessity of returning to an end depot. The NOP introduces a new challenge for environments with obstacles and waypoints, as the agent must provide collision-free path planning-like solutions that visit several waypoints to ultimately achieve a high-level objective.

Our main contribution is the design of a novel deep reinforcement learning-based navigation system, called NaviFormer, that solves the holistic NOP in real-time. NaviFormer approaches each individual subproblem (route planning and path planning) in a global manner, allowing better understanding of the environment and enhancing the quality of solutions. It is based on a Transformer neural network trained with deep reinforcement learning (DRL) to solve the NOP. This Transformer-based network is composed of an encoder-decoder architecture capable of encoding graphs of nodes/waypoints, and combining them with information from the static obstacles of the environment using a special combined attention operation. The resulting embedding, containing global information from the environment, is used to decode the path that maximizes the global performance of the system. Thus, the encoder generates a semantic representation of a scenario by computing a node-graph embedding, while the decoder iteratively predicts, at each discretized time step, the next waypoint to visit and the next direction to follow, allowing the agent to avoid obstacles.

## 2 RELATED WORKS

The existing related works address the navigation problem as two independent subproblems: route and path planning. As a consequence, this section focuses on works related to these topics.

**Route planning** Route planning approaches can be divided into three main categories: linear optimizers, (meta) heuristic algorithms, and machine learning. Linear optimizers are computationally expensive algorithms that seek the optimal solution within a defined set of feasible solutions, subject to linear constraints. Examples include cutting planes, especially for multiple agents Bianchessi et al. (2018); Sundar et al. (2022), and commercial solvers such as OR-Tools Perron & Furnon (2023), Gurobi Gurobi Optimization, LLC (2023), CPLEX Cplex (2022), and others. These methods are generally accurate but have computational requirements that prevent their use in real-time.

To reduce computation time, heuristics Purkayastha et al. (2020) and metaheuristics Dixit et al. (2019); Rahman et al. (2021) are frequently applied. These methods sacrifice some precision in favor of finding approximate solutions within a reduced amount of time. Some approaches include Variable Neighborhood Search (VNS) Bezerra et al. (2023); Pěnička et al. (2019), or Greedy Randomized Adaptive Search Procedure (GRASP) Campos et al. (2014); Júnior & Guimarães (2019). Some bioinspired learning algorithms, which usually fall into the category of metaheuristics, can be applied to routing problems, such as Genetic Algorithms (GA) Mansfield et al. (2021); Xiao et al. (2022), Particle Swarm Optimization (PSO) Islam et al. (2021); Xiao et al. (2022), Ant Colony Optimization (ACO) Wang et al. (2020); Xiao et al. (2022), and Gray Wolf Optimization (GWO) Li et al. (2021a); Panwar & Deep (2021). Although (meta) heuristic methods are good alternatives to improve computation speed of linear optimizers, they still tend to fail to reach real-time performance.

Machine learning solutions primarily rely on neural networks trained with annotated data (supervised learning) or without it (unsupervised and reinforcement learning) to perform specific tasks. Reinforcement learning frameworks, especially DRL frameworks, are particularly interesting for routing problems, as they employ neural networks that learn from the experience by trying actions and receiving rewards from the environment. Unlike linear optimization and heuristic algorithms, which refine solutions through iterative processes within the feasible solution space, DRL methods first encode data from the environment and then yield predictions that maximize the received reward. Current works use different architectures, such as Convolutional Neural Networks (CNN) Jung et al. (2023) or Graph Neural Networks (GNN) Ma et al. (2020), to encode the graph of nodes that represent routing problems. For the decoding step, responsible for sequentially predicting routes, some works have proposed to combine Recurrent Neural Networks (RNN) and Attention models Bahdanau et al. (2016) to obtain Pointer Networks (PN) Vinyals et al. (2015); Bello et al. (2016); Ma et al. (2020). However, recent Transformer networks Vaswani et al. (2017) and their adaptation to routing problems Gama & Fernandes (2020); Kool et al. (2019) have shown greater superiority over PN and other RNN-based methods, due to their faster parallel-like procedure of processing data, and the robustness of multi-head attention encoding, which is more effective for long sequences of data.

**Path Planning** Path planning works can also be classified into the same three categories used for route planning. However, real-time performance becomes more critical due to obstacle avoidance constraints, which are more closely aligned with real-world applications. Thus, even if linear optimizers are used in some cases, like cutting planes Lam et al. (2022), they are not popular; and other types of approach share the same problem, such as brute-force algorithms Sharma et al. (2017).

Graph search algorithms, such as Informed RRT* Mashayekhi et al. (2020), A* Mandloi et al. (2021), D* Ravankar et al. (2017), and D* Lite Jin et al. (2023), exploit heuristic and sampling techniques to accelerate convergence. Informed RRT* integrates RRT Mthabela et al. (2021) and RRT* Chen & Wang (2022) to form a connected node tree with an enhanced heuristic. A* uses admissible heuristics to guide Dijkstra's graph search Luo et al. (2020) and find optimal solutions. D* and D* Lite improve A* speed and dynamic obstacle handling. Other sampling strategies include Probabilistic Roadmaps (PRM) Fei et al. (2019), which constructs a roadmap based on collision-free path probabilities, and Artificial Potential Fields (APF) Pan et al. (2022), which uses attractive and repulsive forces for navigation. Furthermore, bioinspired learning methods, such as GA Zhang et al. (2023), ACO Wu et al. (2023), and PSO Yu et al. (2022), also use (meta) heuristics for path planning.

Some of the mentioned heuristic and sampling-based approaches perform very fast but yield approximate and suboptimal solutions with limited performance. Instead, machine learning methods, especially those focused on neural networks and DRL, have the potential to learn representations of the environment to find better solutions. They often incorporate a CNN to encode binary global maps (representing the whole scenario) and then make predictions through a set of dense layers Liu et al. (2020); Loquercio et al. (2018). Some methods Li et al. (2020; 2021b) extend this paradigm to multi-agent cases by incorporating a GNN to infer a graph embedding from the state of the agents while maintaining the core CNN + dense layers structure. Alternatively, Transformers, and especially Vision Transformers (ViT) Dosovitskiy et al. (2021), have led to the replacement of CNN encoders with more powerful ViT encoders Chen et al. (2022; 2023). In contrast, our approach proposes to dynamically encode reduced local maps representing the agent's surroundings by a lightweight CNN, enabling fast and efficient predictions. Moreover, the proposed framework jointly addresses the interrelated problems of route and path planning, reaching a superior performance that those works that adopt the simpler approach of decomposing the problem into two independent problems.

## 3 NAVIGATION ORIENTEERING PROBLEM FORMULATION

Consider a set of nodes $G = \{0, ..., n+1\}$ representing the visitable regions, where the nodes $0$ and $n+1$ are the start and end depots. An agent is expected to visit the nodes and return to the end depot within a time limit $T$. The NOP seeks routes that are rewarded for visiting regions of $G$ with a set of rewards $R = \{r_0, ..., r_{n+1}\}$ while avoiding a set of $b$ obstacles $O = \{o_i | i = 0, ..., b\}$ represented as circles, where the $i^{th}$ obstacle is parametrized by its center and radious as $o_i = (x^{obs}, y^{obs}, r^{obs})$.

The path followed is discretized with a step length of $t_s$, such that the total number of steps allowed is $L = \left\lfloor \frac{T}{t_s} \right\rfloor$. The objective of the NOP problem is to maximize the constrained function of Eq. 1.

$$\max \sum_{i=0}^{n} \sum_{j=1}^{n+1} r_j \Phi_{ij}, \qquad \Phi_{ij} = \begin{cases} 1 & \text{if } i \text{ was visited inmediately after } j \\ 0 & \text{otherwise} \end{cases} \tag{1}$$

$$\sum_{j=1}^{n+1} \Phi_{0j} = 1 \tag{2} \qquad \sum_{i=0}^{n} \Phi_{i(n+1)} = 1 \tag{3}$$

$$\sum_{i=1}^{n} \Phi_{ij} \in \{0,1\}; \quad j \in \{1,...,n\} \tag{4} \qquad \sum_{j=1}^{n} \Phi_{ij} \in \{0,1\}; \quad i \in \{1,...,n\} \tag{5}$$

$$\sum_{i=0}^{n+1} \Phi_{ii} = 0 \tag{6} \qquad L_{0(n+1)} \leq \sum_{i=0}^{n} \sum_{j=0}^{n+1} L_{ij}\Phi_{ij} \leq L \tag{7}$$

$$L_{ij} = \min_{\nu \in \mathcal{V}} d_{i,j}(\nu); \quad i,j \in G \tag{8} \qquad u_i - u_j + n\Phi_{ij} \leq n-1; \quad i,j \in G \tag{9}$$

Constraints of Eqs. 2 and 3 force the agent to start and finish the paths at regions $0$ and $n+1$, respectively. Eqs. 4 and 5 ensure continuous routes and prevent revisiting nodes. Eq. 6 forbids immediate node revisits. Eq. 7 limits the distance/time budget $T$ (we assume that the agent moves at a constant speed, meaning that the time limit can be converted to a distance limit), which is discretized as $L$, and imposes that this budget allows at least to travel from node $0$ to node $n+1$ ($L_{0(n+1)}$). Restriction of Eq. 8 minimizes path length between nodes ($L_{i,j}$), where $d_{i,j}(\nu)$ is the length of the path $\nu \in \mathcal{V}$ connecting $i$ and $j$. Finally, subtours Miller et al. (1960) are avoided thanks to the constraint of Eq. 9, where $u_i, u_j \in \{1,...,n\}$ are the positional order of $i$ and $j$ on the path.

## 4 NAVIFORMER NEURAL NETWORK

Since the NOP imposes to find a path that maximizes the visited regions, we let our network predict action $a_t^d \in D = \{0, \frac{\pi}{2}, \pi, \frac{3\pi}{2}\}$ indicating the direction the agent should follow at each time step $t \in \{1,...,L\}$. These prediction of these directions should construct a trajectory that could solve the problem. However, we wanted to improve the network insight about the scenario by letting it also predict each next goal to visit $a_t^g \in G$. Note that actions $a_t^d$ and $a_t^g$ are used to solve Eqs. 1 and 8.

Therefore, NaviFormer, depicted in Figure 2, is based on an encoder-decoder Transformer architecture that first encodes the regions to travel to, characterized by their position ($x_i, y_i; i \in G$) and reward ($r_i$), and the obstacles to avoid, both as a graph of nodes ($h^{graph}$). This embedding is a projection into a $\eta$-dimensional feature space to extract relevant and discriminating information. Then, it uses that information to predict a policy $\pi_\theta(a_t^g, a_t^d|s_t)$ that represents the probability distribution of possible actions $a_t^g, a_t^d$ on each state $s_t$. The prediction of this policy also depends on two additional modules: a state embedding ($h_t^{state}$), with information about the spatial position of the agent and the elapsed time, combined with $h^{graph}$ to find the best next goal ($\pi_\theta(a_t^g|s_t)$); and a direction predictor, where a set of local maps allows predicting the best direction ($\pi_\theta(a_t^d|s_t)$). In this manner, our approach can infer efficient solutions restricted by the problem constraints mentioned above.

**Encoder** NaviFormer encoder, based on standard route planning Transformer encoders like Kool et al. (2019); Sankaran et al. (2023); Fuertes et al. (2023), receives individual linear projections of the input nodes $h^{lin}$ and obstacles $h^{obs}$ with dimension $\eta = 128$, and generates a combined graph embedding $h^{graph}$ by learning some attention scores $S$ that promote those node connections that improve the expected reward in the long-term. Contrary to standard encoders that combine pairs of input data, NaviFormer encoder considers a three-way relationship to find the affinity between every pair of nodes with respect to each obstacle. For that purpose, the combined multi-head attention strategy of Figure 3a is proposed to substitute standard self-attention mechanisms of Transformers. It takes the feature vectors $h^{lin}$ and $h^{obs}$, and applies a self-attention operation to find the affinity between nodes from the same set, and an attention operation between the two embeddings $h^{lin}$ and $h^{obs}$ to find the crossed-affinity. The output of both operations ($h_{11}$ and $h_{12}$ from $h^{lin}$, and $h_{21}$

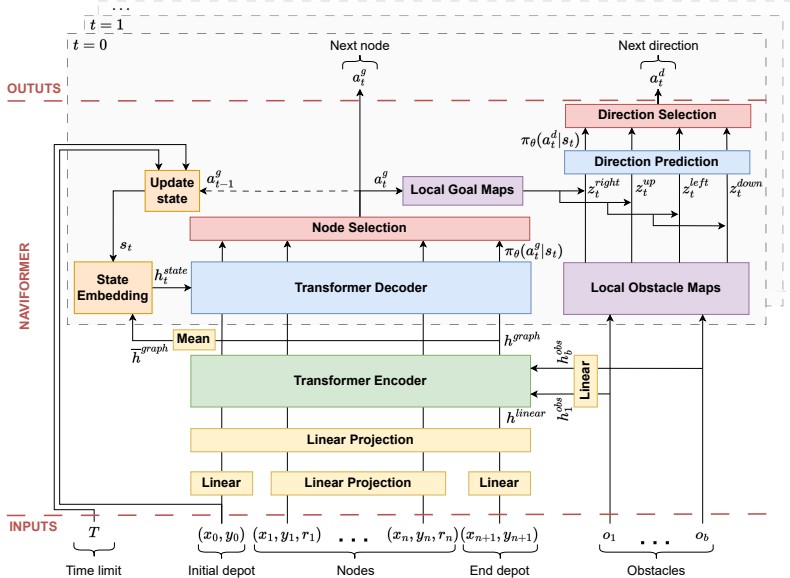

Figure 2: Architecture of NaviFormer, which receives information from the environment (depots, regions, reward per region, obstacles, time limit) and predicts a policy $\pi_\theta(a_t^g, a_t^d|s_t)$.

and $h_{22}$ from $h^{obs}$) are merged through another pair of attention layers to obtain the final combined embeddings $h_1$ and $h_2$. This mechanism can be expressed as follows.

$$h_{11} = \text{MHA}(h^{lin}, h^{lin}, h^{lin}); \; h_{12} = \text{MHA}(h^{lin}, h^{obs}, h^{obs}); \; h_1 = \text{MHA}(h_{11}, h_{12}, h_{12}) \quad (10)$$

$$h_{21} = \text{MHA}(h^{obs}, h^{lin}, h^{lin}); \; h_{22} = \text{MHA}(h^{obs}, h^{obs}, h^{obs}); \; h_2 = \text{MHA}(h_{22}, h_{21}, h_{21})$$

where $\text{MHA}(h^Q, h^K, h^V)$ is the multi-head attention operation. Notice the difference between self-attention (same embedding for query, key, and value), and (crossed) attention (different embeddings). The encoding block of Figure 7a in Appendix A is stacked $N = 3$ times to obtain a deeper model. For the last block, $h_2$ is not necessary since the resulting $h^{graph}$ is just inferred from $h_1$.

**State Embedding** In addition to the scenario encoding, the agent's state $s_t$ is also encoded by the state embedding module (see Figure 3b). It includes the agent's position $c_t$ and time $t_t^{left}$, the distance to the obstacles $o_1, ..., o_b$, and the agent's provenance (included in the graph embedding of the last node $h_{a_{t-1}^g}^{graph}$). The final state embedding ($h_t^{state}$) is obtained by adding the linear projection of the aforementioned elements and the linear projection of the averaged graph embedding across all nodes $\overline{h}^{graph}$ that provides some context about the agent's location on the graph.

**Decoder** Unlike the encoder, which generates a unique static $h^{graph}$, the decoder (Figure 7b in Appendix A) predicts actions at every time step from the scenario information $h^{graph}$ and the agent's state $h_t^{state}$. The first decoding module, the masked multi-head attention, combines both inputs with an attention layer that includes a mask $M$ to satisfy specific constraints of Section 3 as:

$$S(h_t^{state}, h^{graph}) = \text{SoftMax}\left(\frac{M\left(Q(h_t^{state})K(h^{graph})^T\right)}{\sqrt{\eta}}\right), \quad (11)$$

$$M(h_i) = \begin{cases} -\infty & \text{if } i \text{ was visited} \\ h_i & \text{otherwise} \end{cases}, \forall i \in G \quad (12)$$

This mask ensures that no region is visited multiple times. The last module of the decoder is a masked single-head attention that uses a tanh activation function and a unique head to predict a multinomial probability distribution $\pi(a_t^g|s_t)$, from which the next node $a_t^g$ is sampled.

**Direction Prediction** In addition to $\pi(a_t^g|s_t)$, NaviFormer estimates another distribution $\pi_\theta(a_t^d|s_t)$ to obtain a prediction of the agent's direction. For this purpose, local maps (see Figure 3c) centered

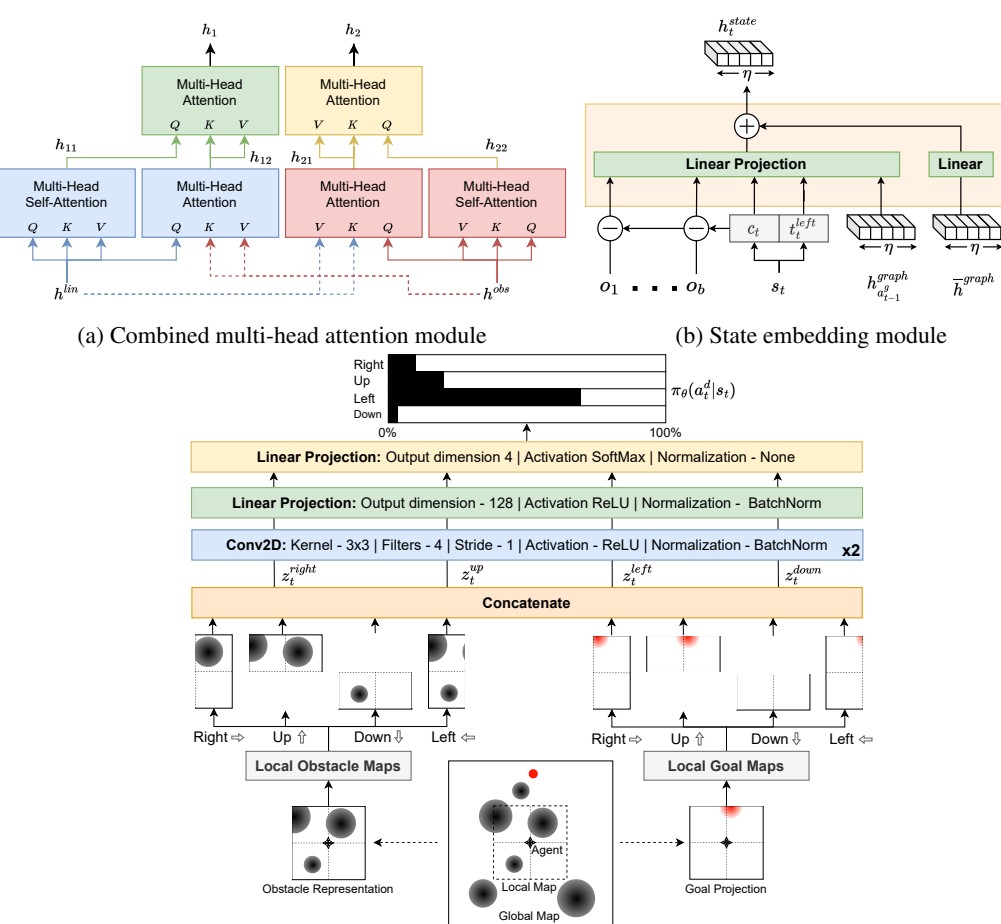

(a) Combined multi-head attention module  (b) State embedding module

(c) Direction prediction layers

Figure 3: Several novel NaviFormer modules: (a) the combined multi-head attention operation to merge node and obstacle information, (b) the state embedding module to encode the situation of the agent at each time step $t$, and (c) the direction prediction layers to predict the policy $\pi_\theta(a_t^d|s_t)$ from local maps based on the obstacles and the next goal $a_t^g$.

at the agent's position are generated. The local obstacle map along with the next selected node (projected to appear on the maps) are divided into four sections representing the agent's north ($z_t^{up}$), east ($z_t^{right}$), south ($z_t^{down}$), and west ($z_t^{left}$). These maps sections provide knowledge of the scenario to predict the best direction to reach the next node while avoiding obstacles. Finally, the policy $\pi_\theta(a_t^d|s_t)$ is obtained by processing the local maps by a couple of convolutional and dense layers.

**Training Strategy** NaviFormer is trained with DRL by simulating episodes of different scenarios or problem instances $\alpha$ and collecting, for each $\alpha$, the following reward values:

$$r^{\pi_\theta}(\alpha) = \sum_{a^g \in A^g} \gamma \frac{r_{a^g}}{n/2} - \beta \sum_{a^d \in A_{a^g}^d} d(c_t, a^g) + \xi, \quad \xi = \begin{cases} +20 & \text{if episode is succesful} \\ -10 & \text{otherwise} \end{cases} \tag{13}$$

where $r^{\pi_\theta}(\alpha)$ is the reward received after following $\pi_\theta$ in $\alpha$, $A^g$ is the set of nodes visited, $A_{a^g}^d$ is the set of directions followed to reach $a^g$, $d(c_t, a^g)$ is the distance between $c_t$ and $a^g$, and $\gamma = 10$, $\beta = 0.3$ are constant values. Besides, the reward for visiting each region is $r_{a^g} = 1$ if $a^g \in \{1, ..., n\}$ and 0 otherwise. To maximize the reward collection for the NOP (Eq. 1), we extend Reinforce Williams (1992) to a vanilla Actor-Critic by using a critic value $V^{\pi_\theta}(\alpha)$ as baseline $b(\alpha)$, which

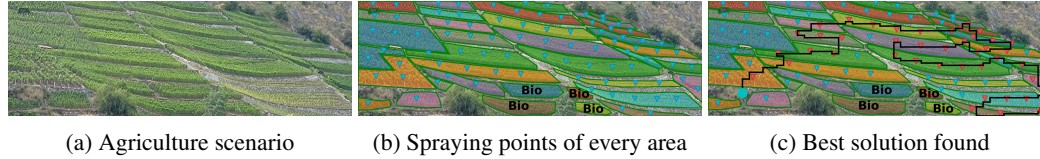

(a) Agriculture scenario     (b) Spraying points of every area     (c) Best solution found

Figure 4: A scenario with (a) cultivation and biocultivation areas, (b) their segmentation and spraying points (triangles), and (c) NaviFormer's solution from start to end depot (cyan and red circles).

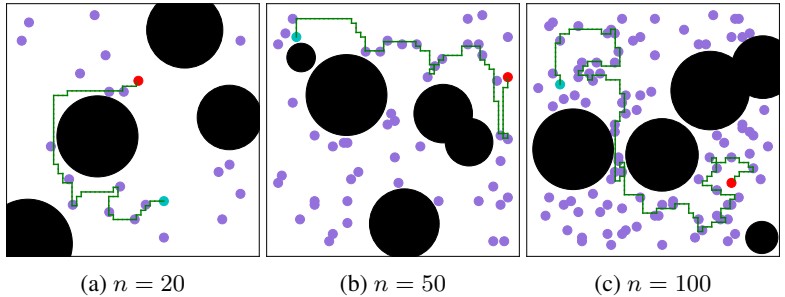

(a) $n = 20$       (b) $n = 50$       (c) $n = 100$

Figure 5: Solutions provided by NaviFormer for some synthetic scenarios.

reduces the variance of the obtained reward across different $\alpha$ and improves the learning process. The gradient of the resulting loss function $\mathcal{L}(\theta|\alpha)$ for the gradient descent update is defined below.

$$\nabla\mathcal{L}(\theta|\alpha) = E_{\pi_\theta(\nu|\alpha)}\left[(r^{\pi_\theta}(\alpha) - b(\alpha))\nabla\log\pi_\theta(\nu|\alpha)\right], \qquad b(\alpha) = V^{\pi_\theta}(\alpha) \qquad (14)$$

where $\nu$ is the set of actions $a = (a^g, a^d)$ that define a path, and $V^{\pi_\theta}(\alpha)$ is predicted from $\overline{h}^{graph}$ by applying two dense layers of hidden size $\eta = 128$ connected through a ReLU activation function. Additionally, to encourage the exploration of new actions during training, we sample them from the binomial distribution $\pi_\theta(a_t|s_t)$ so that the best known actions are more likely to be exploited, but other actions could also be explored. During inference, greedy selection is performed instead.

## 5 RESULTS AND DISCUSSIONS

NaviFormer (code is publicly available [1]) has been evaluated with both real and synthetic data. Synthetic data comprises 640k train samples, 10k validation samples, and 10k test samples. Train and validation sets contain between 10 and 100 visitable nodes with random variation, while the test set covers small (20 nodes), medium (50 nodes), and large (100 nodes) scenarios. Node coordinates were sampled from a uniform distribution $\mathcal{U}(0, 1)$. A random obstacle number from 0 to 5 is also sampled at random locations with radius $r^{obs} \sim \mathcal{U}(0.05, 0.2)$. The time limit $T$ was set to allow visiting around half of the nodes, since these cases tend to be more challenging Vansteenwegen et al. (2009), resulting in $T = 2, 3, 4$ for $n = 20, 50, 100$. The time step remained constant at $t_s = 0.02$.

Real data from a real-world application about pesticide spraying from an unmanned aerial vehicle (UAV) has also been used. This application includes two different types of areas to consider: cultivation and biocultivation (pesticide-free) areas (see Figure 4). The UAV should cover the first ones with pesticide, and completely avoid flying over the latter ones. To fully cover each cultivation area, they are discretized into multiple spraying points, which are considered as nodes for the navigation problem. Moreover, the most restrictive constraint between the UAV's pesticide capacity and fuel is used for its return to the depot. We adapted PASTIS dataset Garnot & Landrieu (2021), originally designed for image segmentation, for this task by normalizing the coordinates in the range [0, 1], setting $T$ similarly to the synthetic data, and randomly choosing biocultivation areas (ranging from 0 to 5) as obstacles (enclosed in circles like those from the synthetic data). The resulting number of PASTIS's instances is 2344, with an average of 42 nodes per instance.

---

[1] Anonymus URL

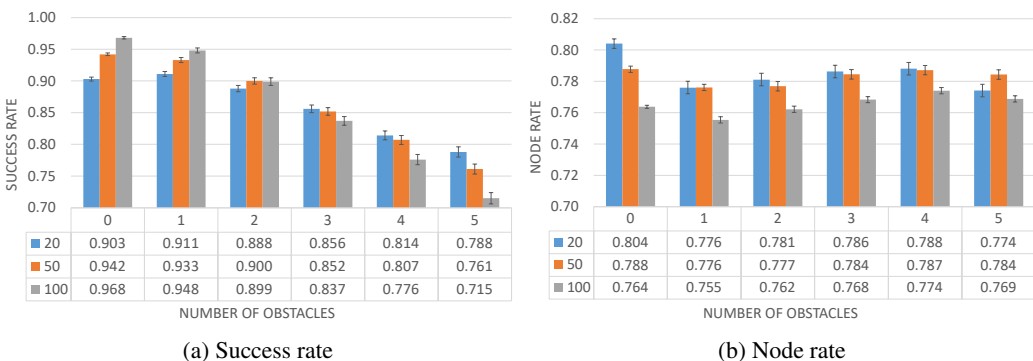

|  | 0 | 1 | 2 | 3 | 4 | 5 |
|---|---|---|---|---|---|---|
| 20 | 0.903 | 0.911 | 0.888 | 0.856 | 0.814 | 0.788 |
| 50 | 0.942 | 0.933 | 0.900 | 0.852 | 0.807 | 0.761 |
| 100 | 0.968 | 0.948 | 0.899 | 0.837 | 0.776 | 0.715 |

NUMBER OF OBSTACLES

|  | 0 | 1 | 2 | 3 | 4 | 5 |
|---|---|---|---|---|---|---|
| 20 | 0.804 | 0.776 | 0.781 | 0.786 | 0.788 | 0.774 |
| 50 | 0.788 | 0.776 | 0.777 | 0.784 | 0.787 | 0.784 |
| 100 | 0.764 | 0.755 | 0.762 | 0.768 | 0.774 | 0.769 |

NUMBER OF OBSTACLES

(a) Success rate (b) Node rate

Figure 6: System performance for small, medium, and large ($n = 20, 50, 100$) synthetic scenarios.

**Results** NaviFormer's performance, measured in terms of success rate (rate of episodes that reach the depot in time without colliding) and node rate (nodes visited over expected nodes), is shown in Figure 6 as a function of the number of nodes and obstacles for synthetic test scenarios. As expected, success rates decline with more obstacles, whereas node rates are higher with 0 and 4 obstacles. The first is intuitive, as obstacle-free scenarios are simpler. The latter is attributed to reduced node-to-node distances with abundant obstacles. Comparing small, medium, and large scenarios, node rates are lower for larger ones due to their complexity; and success rates decrease faster with more obstacles for the same reason. Note that NaviFormer was trained with scenarios of a variable number of regions, achieved by adding dummy nodes, to enhance its flexibility and generalization capability. Regarding qualitative results, Figure 4c shows the performance of NaviFormer on a set of cultivation areas, demonstrating its ability to visit 41 out of 85 spraying points. Besides, Figure 5 exposes the solutions found by NaviFormer for different synthetic scenarios.

**Ablation Study** To analyze NaviFormer's contributions compared to other Transformer-based works Kool et al. (2019); Fuertes et al. (2023); Sankaran et al. (2023), the ablation study of Table 1 was carried out by removing or substituting the combined attention encoder, direction prediction layers, and local maps, as well as a comparison with a 2-step NaviFormer approach. All models were trained and tested on medium-sized synthetic scenarios ($n = 50, T = 3$) and computation times were measured on both GPU ($2 \times$ Nvidia Titan Xp) and CPU (Intel Core i9-7900X 3.30GHz).

First, we conducted an experiment to confirm that the proposed joint NaviFormer network (simultaneously acts as route and path planner) performs better than training both components in isolation. We pretrained the base Transformer network included in NaviFormer to predict routes for the OP. Later, this model was employed to fit the direction prediction layers for the navigation task. The results confirm a slight deterioration in both success and note rates, which suggests that the behavior of the route planner is influenced by the path planner and viceversa. Besides, our 1-step approach is end-to-end trainable. Secondly, the combined attention encoder is assesed by evaluating the model with and without a traditional Transformer encoder. Without encoder (just the initial linear projections of Figure 2 are utilized to process the inputs in this ablation test), learning node positions and relationships is difficult, resulting in a poor node rate. Instead, the standard encoder increased the node rate at the expense of success rate. This traditional encoder receives the inputs and obstacles separately to generate two embeddings. The node embedding is utilized by the decoder to make predictions, and it is combined with the obstacle embedding at the state embedding module to provide some insight about nodes and obstacles. The results of this approach are relatively positive, but it relies on the simplicity of the scenarios with small number of obstacles to mantain high success rates. We believe that the success rate would decrease in more complicated scenarios with more obstacle density. Third, different convolutional layers and linear projections have been removed from the direction prediction module, leading to shallower models whose results are worse than the proposed deeper model. Lastly, the importance of local maps to predict the direction is analyzed. The local maps are limited to represent the surroundings of the agent, while the assessed global maps contain the entire scenario. The inclusion of large global maps provides poor performance due to the large compression of the map by the convolutional layers, and increases the computational cost too much. On the other hand, the removal of maps deteriorates accuracy in exchange of faster performance. Regarding general computation time along the presented ablation tests, removing layers

Table 1: Ablation study showing the performance for synthetic scenarios with $n = 50$ and $T = 3$.

| Ablation study (time in ms) | Success rate | Node rate | Time (GPU) | Time (CPU) |
|---|---|---|---|---|
| **NaviFormer** | **0.906±0.006** | **0.820±0.002** | **0.517±0.001** | **3.517±0.011** |
| NaviFormer (2-step) | 0.877±0.007 | 0.814±0.002 | 0.883±0.001 | 3.104±0.003 |
| w/ standard Transf. encoder | 0.863±0.007 | 0.853±0.002 | 0.492±0.001 | 3.203±0.005 |
| w/o encoder (linear layers) | 0.939±0.005 | 0.040±0.000 | 0.480±0.001 | 3.167±0.005 |
| w/ 1 conv. and 1 linear layer | 0.903±0.006 | 0.809±0.002 | 0.491±0.001 | 3.434±0.005 |
| w/ only 1 conv. layer | 0.885±0.007 | 0.833±0.002 | 0.445±0.001 | 3.181±0.005 |
| w/ only 1 linear layer | 0.870±0.006 | 0.799±0.002 | 0.454±0.001 | 3.291±0.004 |
| w/ global maps | 0.642±0.010 | 0.596±0.003 | 0.979±0.001 | 17.105±0.046 |
| w/o maps | 0.771±0.008 | 0.040±0.000 | 0.301±0.001 | 2.294±0.009 |

Table 2: Comparison with some baselines on the PASTIS dataset.

| Algorithms | Success rate | Node rate | Time (GPU) | Time (CPU) |
|---|---|---|---|---|
| **NaviFormer** | **0.991±0.002** | **0.703±0.007** | **0.517±0.001** | **3.517±0.011** |
| NaviFormer (2-step) | 0.980±0.003 | 0.688±0.007 | 0.883±0.001 | 3.104±0.003 |
| OR-Tools + A* | 0.925±0.011 | 0.519±0.081 | - | 10.264±0.346 |
| OR-Tools (5s) + A* | 0.944±0.010 | 0.689±0.008 | - | 3127.292±63.088 |
| GA + A* | 0.958±0.008 | 0.825±0.008 | - | 4246.492±352.139 |
| OR-Tools + D* | 0.934±0.010 | 0.537±0.008 | - | 383.407±5.707 |
| OR-Tools (5s) + D* | 0.939±0.010 | 0.729±0.009 | - | 3644.377±66.864 |
| GA + D* | 0.942±0.010 | 0.902±0.008 | - | 4421.613±340.739 |

generally improved prediction speed, though NaviFormer already performed in real-time. Note that the reported computation time indicates the average time for solving a problem instance.

**Comparison** NaviFormer is compared to some 2-step baselines on the PASTIS dataset, including combinations of A* and D* with GA and OR-Tools (with and without a 5 second guided local search). To compensate their suboptimal performance, we let path planners 8 motion actions (instead of 4) and reduced $T$ by a small $\epsilon = \frac{T}{10}$ to aid obstacle-agnostic route planners. Table 2 highlights NaviFormer's performance that achieves the highest success rate, proving the importance of tackling the problem holistically. In terms of node rate, NaviFormer competes effectively with all methods, only overcomed by GA + A* and D*, whose computational cost is comparatively huge (several orders of magnitude slower). This is partly due to NaviFormer's local maps configuration, which limits the performance by allowing only 4 motion actions, ensuring successful end depot arrivals at the expense of node visits. This is influenced by the high reward value for successful episodes in Eq. 13, helping the agent's learning through successful finishes, but indirectly affecting node rates. Despite that, NaviFormer remains significantly faster than GA and more accurate than OR-Tools, achieving an optimal balance between accuracy and computation speed for real-time applications. In addition, the 2-step NaviFormer is also included in Table 2. As before, our proposal outperforms this variant in terms of accuracy, which confirms the slight improvement in another (real) dataset.

## 6 CONCLUSIONS

In this paper, a novel DRL approach called NaviFormer that solves the holistic navigation (orienteering) problem in real-time is proposed. It combines route planning (waypoint sequencing) and path planning (shortest trajectory prediction) using a Transformer network that includes a novel encoder to efficiently create joint embeddings for waypoints and obstacles, allowing the prediction of next waypoints to visit and safe (non-colliding) directions to reach them. Compared to 2-step state-of-the-art methods, NaviFormer achieves great balance between accuracy and computation time, making it suitable for real-time applications. Future research may focus on improving direction prediction for more motion actions or continuous directions, potentially enhancing visitation rates. Besides, local maps for obstacle avoidance could limit performance in complex scenarios (e.g., mazes).

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

## A  TRANSFORMER DETAILS

Figure 7 shows in detail the structure of some standard modules from NaviFormer (see Figure 2). The encoder (see 7a is based on other route planning works such as Kool et al. (2019); Sankaran et al. (2023); Fuertes et al. (2023). It receives as input the linear projections of nodes and obstacles, and

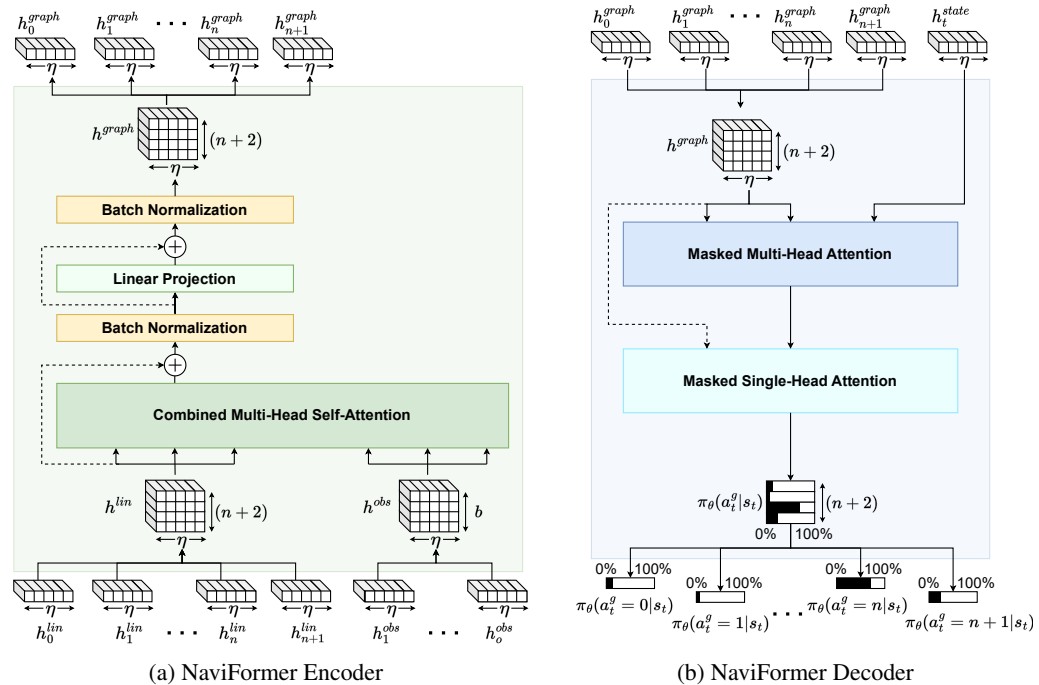

| (a) NaviFormer Encoder | (b) NaviFormer Decoder |
|---|---|

Figure 7: NaviFormer network is mainly composed of (a) an encoder and (b) a decoder. The encoder presents (c) a novel combined multi-head attention operation. The decoder receives an embedding from (d) the state embedding module that encodes the situation of the agent at each time step $t$.

applied th combined multi-head attention technique to generate proper embeddings. The network learns to generate embeddings that encourage those three-way relationships between each pair of nodes (currently visited node and future node to visit) with respect to obstacles that maximize the long-term reward collection.

Note the difference between the combined multi-head attention layer introduced in Section 4 and the standard multi-head attetntion strategy. Traditional general purpose Transformer encoders Vaswani et al. (2017); Dosovitskiy et al. (2021) are capable of generating several scores $S$ with multi-head self-attention:

$$S(h) = \text{SoftMax}\left(\frac{Q(h)K(h)^T}{\sqrt{\eta}}\right), \quad Q(h) = W^Q h, \quad K(h) = W^K h, \tag{15}$$

where $h$ is an embedding, and $Q$ and $K$ are linear projections of $h$ (known as "query" and "key"). This attention scores $S$ are used to weight the values of the called "value", also a linear projection $V(h) = W^V h$ to obtain a final feature vector that "pays more attention" to those more affine nodes. Our combined multi-head attention takes this basic concept of node matching learning and extends it to a three-way relantionship, such that the obstacle information is integrated within the graph embedding in a more robust and intelligent manner.

The graph embedding output by the combined multi-head attention module is linearly projected to reinforce the captured information. It is also processed by some skip connections and batch normalization layers that add stability.

The final graph embedding output by the encoder is properly analyzed later by the decoder (see Figure7b. This decoder employs a common multi-head attention opertation with a mask that ensures the accomplishment of the restrictions and constraints presented in Section 3, as it is commented along Section 4.

