# OpenReview forum: "NaviFormer: A Deep Reinforcement Learning Transformer-like Model to Holistically Solve the Navigation Problem"
_ICLR.cc/2024/Conference — Submitted to ICLR 2024_

### Official Review · Reviewer_wH3n · 2023-10-31

**Soundness:** 2 fair
**Presentation:** 2 fair
**Contribution:** 2 fair
**Rating:** 6
**Confidence:** 3

**Summary:**

This paper studies the navigation planning problem: the combination of route planning (in terms of prize-collecting for maximizing some benefits from the visited points) and path planning (in terms of avoiding obstacles for finding cost-minimum travel paths) using the transformer-based NN named NaviFormer.

Although the problem definition combines existing problems, and the proposed network seems to be a minor improvement or technical contribution using the transformer block, the design of the training strategy (e.g., rewards) was evaluated in experiments. The experimental results show some improvements in the PASTIS dataset.

**Strengths:**

- Interesting navigation problem combining the two existing (possibly traditional) tasks.
- A transformer-based NN design combining multiple information sources (nodes, start/goals, time limits, obstacles), showing good experimental results via the ablation study.

**Weaknesses:**

- Although the proposed problem is a solid optimization problem, some unclear relation (navigation points of G, spraying points, and local/global maps, particularly with static obstacles) remains, which can be clarified by improving the writing.
- Insufficient experiment explanations, rather than giving some trivial figures: For example, in my opinion, some figures in Fig. 3 are not very important to convey the concept and idea of the paper (just following some basic idea of Transformer), but experiment information is much important as the author tries to tackle a new navigation problem. Fig.6 should be updated for clarity without a 3D plot.
- Fixed training strategy in the main paper (no link or mention of different rewards or learning strategies).

**Questions:**

- Compared with Fig. 1, the possible actions are restricted (only four angles). Can we discuss this point? For example, the learning difficulty when increasing the freedom of angles (e.g., 15 degrees each), the optimization problem, and results (e.g., more profitable routes with restricted times?). In the last sentence of the paper as a future work, could considering the continuous action space make the problem completely different (or some minor differences?)

- I’m not sure why the global map interrupts the performance (as pointed out in Table 1). Could you give some additional insights? In my feeling, as the obstacles seem to be fixed (not dynamic), the global map could help the task from the macroscopic viewpoint. However, for example, the training to consider the map seems challenging.

- What is the effect of spraying points? In Fig. 5, they seem to be derived from segmentation results. Any connections between the spraying points and obstacles (Bio?). Are all instances similar as illustrated in Fig.5, or are they completely different?

- Some minor comments:
    - Quick question: how did the authors design A* and D* for the problem? It is interesting if the authors give some details because, in my feeling, I’m not convinced that A* is time-consuming, as Table 2 says when the obstacle information is explicitly given. (Possibly, I misunderstood some settings).
    - Just a small question: what is an intuitive explanation or visualization of synthetic cases? (in real UAV datasets can be understood through Fig.5).
    - Time limit in Fig. 2 (denoted by $T$), but in the optimization problem, it seems to be $L$.

~~~~

- After the response, some of the above questions have been clarified, and therefore my score was updated.

---

> ### Author Response · Authors · 2023-11-18
> **Response to Reviewer wH3n (1/2)**
>
> ## Q1: Some unclear relationships remain (navigation points of $G$, spraying points, and local/global maps).
>
> A1: A better explanation is offered as follows. The navigation points of $G$ are actually the same than the spraying points for the pesticide spraying task. The cultivation areas are segmented and divided into smaller sub-areas, represented by a single spraying point. Each spraying point must be visited by the UAV to cover a small part of each cultivation area. Once all the spraying points are visited, the cultivation area is completely sprayed. We have included an improved explanation about this topic at the second paragraph of section 5.
>
> Regarding the local and global maps, the local maps show some small insight about the agent’s surroundings. Instead, the global maps contain a representation of the entire scenario. Again, the writing has been improved at the paragraph related to the ablation study, in section 5.
>
>
> ## Q2: Insufficient experiment explanations, rather than giving some trivial figures
>
> A2: As a result of the reviewer’s suggestions, we have moved Figures 3.a and 3.b to Appendix A in order to gain more space to expand the results’ section. More specifically, we have included a comparison with a 2-step approach using the NaviFormer modules to certify the profit of jointly training the route planning and path planning modules. In addition, we have exposed some examples of the qualitative performance of NaviFormer with synthetic data. Moreover, we improved the writing of the results’ section to enhance the comprehension of the explanations and discussions provided.
>
> Regarding Figure 6, it has been updated and now it does not contain any 3D plot.
>
> ## Q3: Fixed training strategy in the main paper (no link or mention of different rewards or learning strategies).
>
> A3: We thank the reviewer for the suggestion, we will consider different training strategies, rewards, etc. in future research works. On the other hand, we employed a basic Actor-Critic strategy to face the deteriorating effect of sample variance during the learning process. The distribution of reward values of the nodes is fixed to a constant value of 1 because no node is preferred over the rest for our pesticide spraying application. However, multiple learning strategies and nodes could be utilized to train NaviFormer and achieve different performance results.
>
> ## Q4: Discussion about freedom of angles and continuous actions.
>
> A4: Effectively, these are very pertinent issues to discuss. Increasing the freedom of angles would make the learning process more challenging because the number of solutions to the problem would increase in an exponential manner. In any case, it would be possible to learn the desired policy at expense of a lengthier training procedure and computational cost (for example, the computation of more local maps would be necessary). On the other hand, predicting continuous actions in a continuous action space would approximate better to the optimal solution. However, this type of solutions usually implies convergence issues during the training phase. This is because learning to find the best path to the next selected node using continuous actions requires much more precision, and the probability of predicting a “bad action” is much higher. As a consequence, we have specified in section 6 this limitation of our algorithm to encourage future research work.
>
> ## Q5: I’m not sure why the global map interrupts the performance in Table 1.
>
> A5: The reviewer is right regarding the expected improvement in performance when considering global maps. However, for this ablation test we only focused on modifying the maps analyzed by NaviFormer, and we kept the rest of layers the same. Therefore, the loss in performance is associated to resizing these global maps to the same size expected by the prediction layers exposed in Figure 4 (Figure 3.c in the new version of the paper). We could increase the size of these maps, but the computational cost would increase significantly. Moreover, as the reviewer has pointed out, training a network that considers global maps is much more challenging and the convergence speed would be slower. And considering that the training techniques were the same for all the ablation tests, it is possible that the test related to the global maps has not converged to a global minimum.

---

> > ### Author Response · Authors · 2023-11-18
> > **Response to Reviewer wH3n (2/2)**
> >
> > ## Q6: What is the effect of spraying points?
> >
> > A6: To completely cover every segmented cultivation area, we have divided them into spraying points. The agent (UAV) should visit each spraying point and spray that part of the cultivation. Once all the spraying points have been visited, the complete cultivation area is sprayed. The spraying points are processed by NaviFormer as input nodes, and they can be visited in any order. Regarding the biocultivation areas (“Bio” in Figure 4 of the new version of the paper) they are enclosed into circles and treated as obstacles to prevent small drops of pesticide from falling and affecting the biocultivations. All these new explanations are now contained in the second paragraph of section 5.
> >
> > ## Q7: I’m not convinced that A* is time-consuming, as Table 2 says.
> >
> > A7: A* is not time-consuming, as the reviewer has appreciated. However, it is important to note that the algorithms shown in Table 2 combine route planners (GA and OR-Tools) with path planners (A* and D*). And therefore, it can be stated that the majority of time consumed by the 2-step approaches that employ A*, is consumed by the route planners. In fact, the combination of OR-Tools and A* achieves the second smaller computation time, only surpassed by NaviFormer, which obtains a comparable computation speed.
> >
> > ## Q8: What is an intuitive visualization of synthetic cases?
> >
> > A8: We have included some synthetic examples showing the qualitative performance of the system in Figure 5.
> >
> > ## Q9: Time limit in Fig. 2 (denoted by T), but in the optimization problem, it seems to be L.
> >
> > A9: Both $T$ and $L$ are related to the time limit. However, as it is indicated in the first paragraph of Section 3, $L=⌊T/t_s ⌋$. This means that $L$ is the number of steps allowed for the given time limit $T$.

---

> > > ### Comment · Reviewer_wH3n · 2023-11-20
> > > **Thank you for your responses.**
> > >
> > > I appreciate the authors' responses and revisions in PDF.
> > >
> > > The answers to Q4, Q5, and Q7 clarified my technical concerns. Further, clarifications by answers to Q1-Q3 and Q6, Q8, and Q9 clarified some difficult parts to read and my misunderstanding. The new figure is interesting for me and I understand the idea by the authors. Therefore, I'll increase my score from 5 to 6.

---

### Official Review · Reviewer_ZnEn · 2023-10-31

**Soundness:** 3 good
**Presentation:** 2 fair
**Contribution:** 2 fair
**Rating:** 6
**Confidence:** 3

**Summary:**

This work proposes NaviFormer, a deep reinforcement model that is able to perform route planning and path planning simultaneously. The method is evaluated on one synthetic and one real dataset, demonstrating superior success rates and planning speed to adapted A* and D* methods.

**Strengths:**

1. The paper proposes a novel way that is able to conduct route planning and path planning simultaneously.
2. The design of the neural architecture is clear and well presented.

**Weaknesses:**

1. The designed encoder that integrates the obstacle information into node embeddings using attention seems not new, which has already been proposed in [R].
2. The algorithm is only evaluated on two datasets, and the synthetic dataset lacks enough details. Besides, 640k training samples vs. 10k testing samplings is not a standardized data split, which may cause overfitting.

[R] Yu, Chenning, and Sicun Gao. "Reducing collision checking for sampling-based motion planning using graph neural networks." Advances in Neural Information Processing Systems 34 (2021): 4274-4289.

**Questions:**

1. It is not clear to me how the constraints described in Sec 3 are reflected in the neural architecture design. I hope the authors can give more clarification.
2. Is it necessary to predict both the future actions and directions? Does the predicted action imply the direction?
3. Does navigation problem aim to connecting the start and goal node by traversing a predefined set of nodes? If so, how are the predefined set of nodes are determined?
3. Can NaviFormer deal with dynamic environment? i.e., the cases where the obstacles are moving.

---

> ### Author Response · Authors · 2023-11-18
> **Response to Reviewer ZnEn (1/2)**
>
> ## Q1: The designed encoder seems not new (already proposed in [R]).
>
> A1: We would like to note that the definition of node used in [R] is different from our definition. In [R], nodes are predefined candidate positions that the agent can visit to follow a specific low-level trajectory that solves the path planning/finding problem. Instead, we solve the holistic navigation problem, which combines high-level routes and low-level paths, which are interrelated problems: the low-level paths affect the optimal strategy for the high-level paths (routes). Example: two close nodes that would seem reasonable to be visited can be separated by an obstacle making them in practice far away from each other. Therefore, in our nomenclature, low-level paths are a sequence of coordinates that an agent has to follow between consecutive nodes (but not a full route that satisfies a high-level purpose: maximize the number of visited nodes under restrictions of time). Comparatively, the high-level routes are a sequence of nodes that the agent has to visit, that is, regions to visit as in the orienteering problem or traveling salesman problem. For the pesticide spraying task, for example, each node is a spraying point (divisions of each cultivation to cover the entire area) where the agent has to perform a specific action (in this case, spray pesticide). And the trajectory to fly from one spraying node to the next one is a set of coordinates, that are not restricted to any fixed node position as in [R]. To better highlight the difference for each type of node definition, we will refer to them as low-level and high-level nodes.
>
> Therefore, while it is true that the mentioned paper proposes a GNN encoder to integrate obstacle and low-level node information, our approach does not consider a predefined set of low-level nodes that the agent should follow (as [R] does). Therefore, our approach (which employs a Transformer encoder instead of a GNN) allows the agent to move freely around the scenario without any fixed low-level node position constraint. Besides, the mentioned GNN has to process a large set of low-level nodes to provide profitable paths, which seems to increase the computational time considerably, as the authors briefly comment in the results section, while our approach is focused on the computational efficiency of the estimated solutions. On the other hand, we just encode the information related to the high-level nodes and the obstacles and let the direction prediction layers to infer the low-level path. We can conclude that both methods have a common objective (integrating node and obstacle information) but follow different approaches to that end. They also address different problems: path planning [R] vs navigation planning (ours, which combines path and route planning).
>
> ## Q2: The algorithm is only evaluated on two datasets, and the synthetic dataset lacks enough details. Besides, 640k training samples vs. 10k testing samplings is not a standardized data split, which may cause overfitting.
>
> A2: In this regard, we have adopted the experimental setup of some well-known papers related to reinforcement learning applied to routing problems, such as [1] or [2], which use a similar data split, and therefore allows a better comparison of the results. In fact, the gap between training and testing samples is even larger in these cases: 1,280k vs 10k. In our humble opinion, in reinforcement learning scenarios, the most important issue to face is sample variance, which may prevent the system from learning properly and converging to good solutions. Therefore, very large training sets are necessary to guarantee good performance.
>
> [1] W. Kool and H. van Hoof and M. Welling, Attention, Learn to Solve Routing Problems! ICLR 2019
>
> [2] Irwan Bello, Hieu Pham, Quoc V. Le, Mohammad Norouzi, & Samy Bengio. (2017). Neural Combinatorial Optimization with Reinforcement Learning.

---

> > ### Author Response · Authors · 2023-11-18
> > **Response to Reviewer ZnEn (2/2)**
> >
> > ## Q3: It is not clear to me how the constraints described in Sec 3 are reflected in the neural architecture design. I hope the authors can give more clarification.
> >
> > A3: Constraints described in section 3 essentially ensure that each node is not visited more than once, and that the agent returns to the end depot within the time limit. Since it is not possible to guess the distance between each pair of nodes in a fast and efficient manner, we cannot ensure that the end depot is reached on time. That is the reason to measure the success rate, which analyzes the number of times the agent bumps into an obstacle or does not reach the end depot on time. However, we can prevent the agent from revisiting nodes by including the mask $M$, detailed in equation 13. This mask will set the logit value corresponding to every visited node to $-\infty$, which is converted to a probability of 0 after the SoftMax operation. And that is the manner to include some of the constraints of section 3 into the neural architecture design. In summary, the neural network along with the training strategy are designed for satisfying the aforementioned restrictions, and depending on the specific restriction it is accomplished in a more implicit way or in a more explicit manner (masking).
> >
> > ## Q4: Is it necessary to predict both the future actions and directions? Does the predicted action imply the direction?
> >
> > A4: Actually, the network predicts two types of actions: next node to visit and next direction to follow, since due to the obstacles, the direction to the next node could not be the same to that to the next waypoint. Therefore, we let the network to choose any visitable node and any direction. Later, we provide proper penalization/reward depending on the distance to the chosen node. Thus, the network learns to predict directions that minimize the distance to the chosen node.
> >
> > ## Q5: Does navigation problem aim to connecting the start and goal node by traversing a predefined set of nodes? If so, how are the predefined set of nodes are determined?
> >
> > A5: Not exactly. The state space is continuous, meaning that the coordinates of the position of the agent are in the range [0, 1]. The action space is discrete since the agent can choose four directions (up, down, right, left). Hence, we count a node as visited once the distance to that node is smaller than $t_s=0.02$.
> >
> > ## Q6: Can NaviFormer deal with dynamic environment? i.e., the cases where the obstacles are moving.
> >
> > A6: The scope of our paper only considers static obstacles. Therefore, as it can be observed in Figure 2, the graph embedding $h^{graph}$ is calculated one single time at the beginning, since the iterative calculations of $h^{graph}$ would not improve the performance in a static scenario. However, for a dynamic environment where obstacles are moving, iteratively calculating $h^{graph}$ would make sense, since each new embedding could indicate the presence of new obstacles or moving obstacles. This change in the network could imply a slight deterioration in computational cost, but it could allow solving the navigation problem with moving obstacles.

---

> > > ### Comment · Reviewer_ZnEn · 2023-11-20
> > > **Thanks for addressing my questions.**
> > >
> > > I have increased my score to 6.

---

### Official Review · Reviewer_yhHi · 2023-10-31

**Soundness:** 3 good
**Presentation:** 2 fair
**Contribution:** 2 fair
**Rating:** 6
**Confidence:** 4

**Summary:**

The paper presents a learning-based approach to a specific navigation task, in which predefined waypoints need to be traversed one by one (travelling-salesman style) and obstacles need to be avoided on the way.
The agent has access to the full system state (waypoints, obstacle positions, its own position), the focus is only on predicting good collision-free routes.

A transformer predicts the next waypoint in a route to follow, and in parallel a convnet predicts the actions that lead to the next waypoint.
The transformer routing network is very similar to existing prior work [1].
The first argued advantage of the solution is that both high-level routing and low-level waypoint-to-waypoint navigation are solved in parallel.
The other is that everything is amortized with deep networks, which is faster than traditional solutions.

The networks are trained to maximize an RL objective, trying to visit as many waypoints on a limited budget while avoiding obstacles.
The algorithm is evaluated in a toy 2D setup with up to 5 circular obstacles and up to 100 waypoints to traverse.
It is also applied to a 2D pesticide spraying problem (PASTIS dataset) of similar complexity to the toy setup, on which it is compared to traditional routing + path planning baselines (not deep), mainly showing improved runtime.

[1] W. Kool and H. van Hoof and M. Welling, Attention, Learn to Solve Routing Problems! ICLR 2019

**Strengths:**

- Solving the routing problem jointly with any movement constraints (here obstacles) is a good idea, as fragmented solutions are bound to suffer from local minima. I am not sure if this particular aspect has been addressed with deep learning before, though.
- Aiming to improve the runtime via amortization is also sane, as already shown in [1] and following works. The overall assumptions thus seem sound to me.
- Ablations of some of the assumptions are included.
- The paper was easy to follow, with some room for improvement in terms of the level of detail (see below).

[1] W. Kool and H. van Hoof and M. Welling, Attention, Learn to Solve Routing Problems! ICLR 2019

**Weaknesses:**

- I see the joint training of the transformer and the network that predicts directions as the main contribution. What I am missing in that regard is an ablation of whether this actually works better than training both components in isolation. Right now it is hard to tell whether the routing patterns are really influenced meaningfully by the path-planning constraints. In that sense, I think the paper would be stronger if it considers a baseline where a routing-only transformer is pretrained like in prior art, and then a direction-prediction network is fit to its predicted goals post-mortem.
- Related to the above, the paper would benefit from representative examples of what navigation runs look like, right now these are virtually missing (there is only one pesticide spraying example). This makes it hard to judge how well the routing and obstacle avoidance work together. Particularly because the considered environments are not that complex.
- The overall idea to use a transformer for routing is directly carried over from prior art (e.g. [1]), one can easily tell by the used notation. Still, most of section 4 is about the transformer design. To count this as a contribution, I find the following points important:
    - The ablation of the transformer (top 2 rows of table 1) is very important, to justify any changes in architecture.
    - It appears that the traditional transformer baseline performs quite well in comparison, which makes me question whether the proposed layer structure (cross-attention, etc.) is really necessary.
    - It is argued that the traditional transformer's success rate is lower because obstacles are not encoded in its embeddings. I don't see why this has to be, can't one provide both the list of waypoints and obstacles as one long vector input? And respectively rely on a more generic architecture, like in [1]?
- The direction-prediction net uses info only from the local neighborhood of the agent (based on figure 4). This makes the path-planning greedy, which may work reasonably in the considered experiments, but I doubt it will be optimal in something like apartment layouts or mazes. This should be highlighted as a limitation.
- In terms of the runtime experiments: is the reported runtime of 3.5ms for one forward pass, or for predicting the whole set of goal waypoints + direction actions? It seems like the reported numbers for the traditional methods are for solving the whole problem.
- Some minor points: presentation-wise, while I appreciate the transparency in the network diagrams, I find these are better suited for an appendix, the information in them could be distilled. I also found the equations in sec. 3 somewhat too verbose, given that they are not used at all in the implementation (e.g. the transformer uses masking to implement most of the constraints).

In summary, I believe the experiments should be more focused on proving that the joint training of the transformer and the path planning network is beneficial, as this appears to be the primary novelty of the research. This and the other issues listed above are currently holding me back. The transformer network itself and the associated runtime benefits were already established in prior works, so I wouldn't count these as novel contributions of this paper.

Edit: slightly improving my score, as some of the above points are clearer after the review responses. As requested, the authors reacted to the first two points above.

[1] W. Kool and H. van Hoof and M. Welling, Attention, Learn to Solve Routing Problems! ICLR 2019

**Questions:**

- In the transformer ablations: what does it mean to not have a transformer encoder, how are the waypoint + obstacle inputs processed then?
- In the case of the traditional transformer, why is it that "obstacles are not encoded in the graph embeddings"?
- How are obstacles represented in the pesticide spraying experiments?
- In the introduction, I wouldn't count A* search as a purely heuristic approach for path-planning. Often A* heuristics happen to be admissible (e.g. in point-to-point navigation like in the paper), which still returns optimal solutions. Would you consider adjusting this? This is also related to my comment about picking directions greedily in the weaknesses section.

---

> ### Author Response · Authors · 2023-11-19
> **Response to Reviewer yhHi (1/2)**
>
> ## Q1: The paper should consider a baseline where a routing-only transformer is pretrained like in prior art, and then a direction-prediction network is fit to its predicted goals post-mortem.
>
> A1: Following the reviewer suggestion, we have conducted an additional ablation test (see Table 1) comparing the performance of our proposed NaviFormer and a 2-step NaviFormer where the route planning submodule (base Transformer) is pretrained and used to fit the path planner (direction prediction layers). It can be observed that our proposal achieves a slight improvement in both success and node rates, which indicates that the route planner is lightly influenced by the path planner and viceversa. Besides, our 1-step approach is end-to-end trainable.
>
> ## Q2: The paper would benefit from representative examples of what navigation runs look like.
>
> A2: Figure 5 now includes several synthetic examples showing the qualitative performance of the system.
>
> ## Q3: To count the transformer design as a contribution, I find the following points important:
>
> ### Q3.1: The ablation of the transformer is very important, to justify any changes in architecture.
>
> A3.1: We agree with the reviewer. For that reason, we have improved the explanations provided for the ablation study, which can be found at answer A3.3 for your convenience.
>
> ### Q3.2: The traditional transformer baseline performs quite well in comparison, which makes me question whether the proposed layer structure is really necessary.
>
> A3.2: As the reviewer has pointed out in question Q4, the proposed scenario is not as complex as other possible cases, such as apartment layouts or mazes. Therefore, the traditional encoder of the ablation study, which combines nodes and obstacles in a more simple and standard manner takes benefits from these scenarios with small number of obstacles to find good solutions to the problem. This fact can be corroborated by examining the success rate, which is higher for the NaviFormer encoder. This means that traditional encoders try to maximize the node rate even at the expense of risking the safe return to the end depot. In comparison, Naviformer is more cautious and tends to sacrifice a little bit the quantity of nodes to visit but ensuring the safe return of the asset. We believe that this policy is generally much more desirable, since collisions tend to be more catastrophic than smaller node rates in real-world applications.
>
> ### Q3.3: Can't one provide both the list of waypoints and obstacles as one long vector input and rely on a more generic architecture?
>
> A3.3: We think the result’s section was not properly explained and therefore understood. We apologize for any misunderstanding and offer better explanation next. The traditional transformer encoder presented in the ablation study in fact performs something similar to what the reviewer comments. This encoder is fed with the nodes to generate a node embedding, and also with the obstacles to generate an obstacle embedding in a separate manner. Later, both (node and obstacle) embeddings are combined in the state embedding module. We have expanded the results section with better explanations about the experiments and, more specifically, about the ablation tests to avoid any misunderstanding.
>
> ## Q4: Local maps may work reasonably in the considered experiments, but I doubt it will be optimal in something like apartment layouts or mazes.
>
> A4: We agree with the reviewer that local maps may not be the best approach for more complex scenarios like apartment layouts or mazes, and it is a limitation that we have highlighted in the conclusions’ section to encourage future research.
>
> ## Q5: Is the reported runtime of 3.5ms for one forward pass, or for predicting the whole set of goal waypoints + direction actions?
>
> A5: The computation time reported for both the ablation study and the comparison with other state-of-the-art algorithms indicates the time required to solve the whole navigation problem to offer a fair comparison with other approaches. It is now indicated at the end of the ablation study, in section 5.
>
> ## Q6: While I appreciate the transparency in the network diagrams, I find these are better suited for an appendix.
>
> A6: Following the recommendation of the reviewer, we have moved Figure 3.a and 3.b to Appendix A.

---

> > ### Author Response · Authors · 2023-11-19
> > **Response to Reviewer yhHi (2/2)**
> >
> > ## Q7: In the transformer ablations: what does it mean to not have a transformer encoder?
> >
> > A7: A better explanation is as follows. Figure 2 shows the set of linear projection layers are used to process both the input nodes and obstacles previous to the Transformer encoder. The mentioned ablation test refers to not using any type of Transformer encoder and to process the input data just with the initial linear projection layers. We understand that it may not be clearly explained in the original version of the paper. For that purpose, Table 1 now indicates “w/o encoder (linear projection)” and the text in the ablation study section has also been updated correspondingly.
> >
> > ## Q8: In the case of the traditional transformer, why is it that "obstacles are not encoded in the graph embeddings"?
> >
> > A8: Traditional route planning-based Transformer encoders, such as [1] or [2], do not consider obstacles as input, only nodes. And the direct inclusion of them is not evident since nodes are represented by points (no area) and obstacles have an area that cannot simply represented by points. In conclusion, they have different structure representations. Since this is an ablation study, we maintain the same architecture of NaviFormer but changing our combined multi-head attention encoder (which encodes nodes and obstacles simultaneously/jointly) by a traditional Transformer encoder (which encodes nodes and obstacles separately/independently, whose embeddings are later merged at the state embedding module to provide some insight about the obstacles). Therefore, this modified network considers the obstacles but in a more standard/simple manner than our combined approach. Besides, the ablation test considering the traditional encoder still maintains the Local Obstacle Maps, which are used to avoid obstacles. Consequently, we have improved the explanation of the ablation experiments to prevent misunderstandings.
> >
> > [1] W. Kool and H. van Hoof and M. Welling, Attention, Learn to Solve Routing Problems! ICLR 2019
> >
> > [2] P. Sankaran, K. McConky, M. Sudit and H. Ortiz-Peña, "GAMMA: Graph Attention Model for Multiple Agents to Solve Team Orienteering Problem With Multiple Depots," in IEEE Transactions on Neural Networks and Learning Systems, vol. 34, no. 11, pp. 9412-9423, Nov. 2023, doi: 10.1109/TNNLS.2022.3159671.
> >
> > ## Q9: How are obstacles represented in the pesticide spraying experiments?
> >
> > A9: Biocultivation areas (“Bio” in Figure 5) are enclosed into circles that represent the obstacles. Agents (UAVs) should avoid flying over these areas to prevent small drops of pesticide from falling and affecting the biocultivations. Second paragraph of section 5 now clearly specifies this information.
> >
> > ## Q10: In the introduction, I wouldn't count A* search as a purely heuristic approach for path-planning. Would you consider adjusting this?
> >
> > A10: We agree with the reviewer. The paper now indicates that A* is a graph search method with admissible heuristics that guides Dijkstra’s graph search to find optimal solutions.

---

> > > ### Comment · Reviewer_yhHi · 2023-11-22
> > >
> > > Thank you for responding to all of my concerns.
> > >
> > > I am willing to slightly increase my score. The new 2-step ablation indicates that the joint training has a slight advantage, full transparency in this regard is important. The added clarifications about the ablations and the new figure are also appreciated.

---

### Official Review · Reviewer_ayad · 2023-11-01

**Soundness:** 3 good
**Presentation:** 3 good
**Contribution:** 2 fair
**Rating:** 6
**Confidence:** 3

**Summary:**

The authors propose a transformer policy network trained using deep reinforcement learning to jointly solve the route planning (visit most number of nodes in minimum time) and path planning (generating collision-free paths to goal) problem. The modified transformer architecture applies multi-head attention between graph node and obstacle embeddings (section 4, page 5).

The authors evaluate performance on synthetic and real-world data (PASTIS dataset). An ablation study is included for synthetic data (Table 1) while a comparison to other approaches is done for the PASTIS dataset (Table 2). The authors find a computation time speed up and success rate increase versus more conventional 2-step routing and path planning methods.

**Strengths:**

- State-of-the-art performance.
- The work is well written and clear.
- Included ablations are useful for analyzing the components of the proposed network.
- Tested on multiple datasets (synthetic and real-world).

**Weaknesses:**

- When comparing to baselines, all approaches appear to be heuristic methods. It would have been useful to include a machine learning 2-step baselines as the improvement could simply be due to the different class of algorithm instead of the authors' assertion that NaviFormer beats baselines due to its tackling the routing and path planning problem jointly.

- How repeatable are the results over different seeds and validation set splits? It would have been useful to include a variance or confidence interval with the reported results.

**Questions:**

- How repeatable are the results over different seeds and validation set splits? It would have been useful to include a variance or confidence interval with the reported results.
- At the beginning of section 4, $a_t^d$ can take the value $\frac{2\pi}{3}$. I assume this should be $\frac{3\pi}{2}$?
- In the ablation study, can you clarify what you mean by the “traditional encoder”? It is written that “obstacles are not encoded in the graph embedding” but I assume that they are still input into the Local Obstacle Maps in Figure 2? That is to say that this approach only has the $h^{\mathrm{obs}}$ connection into the Transformer Encoder in Figure 2 is removed?

__Edit: After rebuttal from the author, I have raised my score slightly.__

---

> ### Author Response · Authors · 2023-11-19
> **Response to Reviewer ayad**
>
> ## Q1: It would have been useful to include machine learning 2-step baselines
>
> A1: We appreciate the reviewer suggestion. In that sense, we have conducted an additional experiment (see Table 1) comparing the performance of our proposed NaviFormer and a 2-step NaviFormer where the route planning submodule (base Transformer) is pretrained and used to fit the path planner (direction prediction layers). It can be observed that our proposal achieves a slight improvement in both success and node rates, which indicates that the route planner is lightly influenced by the path planner and viceversa. Besides, our 1-step approach is end-to-end trainable.
>
> ## Q2: How repeatable are the results over different seeds and validation set splits?
>
> A2: Following the suggestion of the reviewer, we have included the confidence interval with the results reported in Figure 6, Table 1, and Table 2. It can be observed that the intervals are very small, which suggests that results are potentially repeatable.
>
> ## Q3: At the beginning of section 4, $a_t^d$ can take the value $2π/3$. I assume this should be $3π/2$?
>
> A3: The reviewer is right. It was a mistake, but now it is properly written in the paper.
>
> ## Q4: In the ablation study, can you clarify what you mean by the “traditional encoder”?
>
> A4: That is partially correct. On the one hand, traditional route planning-based Transformer encoders, such as [1] or [2], do not consider obstacles as input, only nodes (since it is not trivial to consider them in such approaches and, for that reason, it is a clear contribution of this paper). On the other hand, since this is an ablation study, we maintain the same architecture of NaviFormer but changing our combined multi-head encoder (which encodes nodes and obstacles simultaneously) by a traditional Transformer encoder (which encodes nodes and obstacles separately, whose node embeddings are later used in the decoder merging them into the state embedding module to provide some insight about the obstacles). Therefore, this modified network does consider the obstacles but in a more standard and simple manner than our combined approach. Besides, as the reviewer comments, the ablation test considering the traditional encoder still maintains the Local Obstacle Maps of Figure 2, which are used to predict directions that avoid obstacles. As a consequence of the reviewer’s comments, we have improved the explanation of the ablation experiments to prevent misunderstandings.
>
> [1] W. Kool and H. van Hoof and M. Welling, Attention, Learn to Solve Routing Problems! ICLR 2019
>
> [2] P. Sankaran, K. McConky, M. Sudit and H. Ortiz-Peña, "GAMMA: Graph Attention Model for Multiple Agents to Solve Team Orienteering Problem With Multiple Depots," in IEEE Transactions on Neural Networks and Learning Systems, vol. 34, no. 11, pp. 9412-9423, Nov. 2023, doi: 10.1109/TNNLS.2022.3159671.

---

> ### Comment · Reviewer_ayad · 2023-11-20
>
> Thank you for replying to my feedback. As a follow up to question 1: Can the extra machine learning baseline also be applied to the PASTIS dataset (Table 2) in addition to the synthetic dataset? I was referring to the results in in Table 2 when mentioning a lack of machine learning baselines.

---

> > ### Author Response · Authors · 2023-11-20
> > **2nd Response to Reviewer ayad**
> >
> > Yes, the reviewer is right in the sense that the 2-step NaviFormer approach can be included in both Table 1 and Table 2, achieveing a more valuable comparison for the PASTIS (real) dataset. We have now included that comparison in Table 2.

---

> > > ### Comment · Reviewer_ayad · 2023-11-22
> > >
> > > Thank you for addressing my concerns. I have raised my score slightly.

---

### Meta-Review · Area_Chair_gWj8 · 2023-12-11

**Metareview:**

This paper presents a new transformer model architecture for selecting simultaneously a waypoint and a discrete action to move towards that waypoint to solve route planning and path planning tasks. The central claim is that solving both problems simultaneously allows for better solutions to be found with less computation. The reviewers acknowledge that the paper is generally well done, but its novelty is low. While the results suggest that there may be a speed improvement, it is unclear where this improvement comes from and how reliable it is. The OR-Tools plus A* implementation is only 6 seconds beyond the NaviFormer. Since computation times are often highly dependent on code and pipeline optimization, it is unclear if speed improvement comes from an architectural innovation. The paper also does not indicate how many trials were used and what the +/- terms indicate (standard error, standard deviation, confidence intervals), which prevents accurate interpretation of the results.

Overall, the paper places high emphasis on the performance results. However, it is not clear how better performance is being achieved. This makes it very difficult to identify when this method would be a good choice for other problems. So, I am not recommending this paper for acceptance.

**Justification For Why Not Higher Score:**

This paper could be rated higher as the reviewers were all borderline accept, and no significant objections were made post-rebuttal. My assessment of the reviews is that while the paper did not have many large flaws, it was not exciting, i.e., they did not learn much from the paper. I felt similarly about the paper and did not see a strong reason to accept it. However, I also have no strong reason to reject it.

**Justification For Why Not Lower Score:**

N/A

---

### Decision · Program_Chairs · 2024-01-16

Reject